# Photo Stimulation of Seminal Doses with Red LED Light from Duroc Boars and Resultant Fertility in Iberian Sows

**DOI:** 10.3390/ani11061656

**Published:** 2021-06-02

**Authors:** Sara Crespo, Mateo Martínez, Joaquín Gadea

**Affiliations:** 1Department of Physiology, International Excellence Campus for Higher Education and Research “Campus Mare Nostrum” and Institute for Biomedical Research of Murcia (IMIB-Arrixaca), University of Murcia, 30100 Murcia, Spain; 2Technical Department, CEFU, SA, Alhama de Murcia, 30840 Murcia, Spain; mateo.martinez@cefusa.com

**Keywords:** porcine AI, spermatozoa, reproductive outcomes

## Abstract

**Simple Summary:**

The main objective of breeding Iberian pigs is the production of high-quality dry cured meat products. As this breed shows a reduced litter size in comparison to the commercial breeds, some previous studies have reported the use of photostimulation of seminal doses as a method for improving the farrowing rate and litter size. The aim of this study was to explore whether the application of a photostimulation procedure to Duroc seminal doses has any beneficial effect on fertility and litter size. Semen samples were obtained from 38 fertile Duroc boars and the fertility study was conducted on two commercial farms using multiparous Iberian sows (farm A, n = 824; farm B, n = 2131), that were randomly assigned to LED (L) or control (C) groups. Post-cervical insemination took place 0 and 24 h after the diagnosis of estrus, with seminal doses from the same ejaculate and same treatment. The photostimulation of the seminal doses had no effect on the reproductive performance (farrowing rate: 91.72% C vs. 90.09% L, litter size: 8.71 ± 0.06 piglets C vs. 8.70 ± 0.05 L, *p* > 0.05).

**Abstract:**

In pigs, it has been reported that increased farrowing rates and litter size have been induced by photostimulating the seminal doses for artificial insemination with red LED light. As the reproductive characteristics, production system, and outcome parameters of Iberian breed pigs are different from other commercial breeds, the aim of this study was to evaluate the possible effect of illuminating seminal doses from Duroc boars with red LED light and the fertility outcomes of Iberian females. Semen samples were obtained from 38 fertile Duroc boars. Photostimulation of the artificial insemination (AI) seminal doses was carried out by illuminating the samples with a red LED for 10 min, followed by 10 min of darkness, and finally 10 additional minutes of red light. The fertility study was conducted on two commercial farms using multiparous Iberian sows (farm A, n = 824; farm B, n = 2131), that were randomly assigned to LED (L) or control (C) groups. No differences were found between L and C groups in both farms (*p* > 0.05) for parity, pregnancy rate, duration of pregnancy, farrowing rate, and litter size (total, alive, and stillborn piglets). Farrowing rates in farm A were 88.8% (n = 383) for control and 89.6% (n = 441, *p =* 0.67) for the LED group. In farm B, farrowing rates were C:90.5% (n = 1030) and L: 90.1% (n = 1101, *p =* 0.48). In farm A, total born piglets were 8.69 ± 0.11 for C and 8.71 ± 0.11 for L (*p =* 0.87). In farm B, the results were 8.72 ± 0.7 for C and 8.70 ± 0.06 (*p =* 0.82) for L. Under the production conditions for the Iberian breed, the photostimulation with red LED light using Duroc pig seminal doses was not effective in improving the fertility of Iberian sows.

## 1. Introduction

Previous studies have pointed out the sensitivity of sperm cells to light exposure. Differences between species, type of light, intensity and duration of the illumination used in these studies have produced controversial outcomes [1,2,3]. In pigs, a red LED-based photostimulation procedure (10 min illumination, 10 darkness and 10 illumination) increased the achievement of in vitro capacitation and subsequent progesterone-induced acrosome reaction [4]. In relation to in vivo fertility outcomes in pigs, increased farrowing rates and litter sizes have been reported following photostimulation of the seminal doses for artificial insemination by red LED light [4,5].

The previous studies were developed with commercial breeds in mind, mainly Large White and landrace breeds. However, the reproductive parameters and outcomes of the Iberian breed are different from the commercial breeds [6,7]. Normal practices and regulations related to Iberian pig products dictate that the sow is Iberian, whereas the boar could be Iberian, Duroc, or a hybrid between the two [8]. The Duroc crossing provides increased prolificacy and productive performance [9,10]. According to Casellas et al. [11], in 2019, the litter size in different varieties of Iberian breed ranged from 8.02 for the Entrepelado variety and 8.40 for the Rentito variety, and these values are different from the data reported for the hyper-prolific genetic lines that are currently being bred, with frequent litters of 18–20 piglets [12].

One of the main objectives in the production of the Iberian breed is the improvement in the reproductive outcomes, especially the litter size, while maintaining the special characteristics of the carcass, destined to be processed as high-quality dry cured meat products [13]. Therefore, the aim of this study was to evaluate the possible beneficial effect on reproductive parameters of illuminating seminal doses from Duroc boars with red LED light in their specific production system.

## 2. Materials and Methods

### 2.1. Ethics

The study was developed according to Spanish Policy for Animal Protection (RD 53/2013), which conforms to European Union Directive 2010/63/EU regarding the protection of animals used in scientific experiments. 

### 2.2. Semen Recovery and Seminal Doses Preparation

Semen samples were obtained from 38 fertile, healthy, and sexually mature Duroc boars. The age of the boars was 20.10 ± 1.78 months old, ranging from 10 to 45 months. All boars were housed in two artificial insemination (AI) centers located in Murcia (Spain). The boars were subjected to the same management conditions, specifically, they were housed in individual pens in buildings with a controlled light regime (16 h per day) and temperature (15–25 °C) and with free access to water. The boars were fed with commercial feedstuff according to the nutritional requirements for adult boars subjected to regular ejaculate collection (once or twice per week) [14].

Sperm-rich fractions were collected by an experienced operator using the gloved hand technique [15] and immediately transported to the laboratory. The sperm-rich fraction was diluted with a commercial extender (MR-A, KUBUS SA, Madrid, Spain), split into seminal doses of 80 mL and cooled to 16 °C, with total motility and normal morphology higher than 70%, and 3 × 10^9^ spermatozoa per seminal dose, and stored at 16 °C for up to 48 h before AI. 

For motility evaluation, two subsamples were placed on warm glass slides (38 °C) and examined under a contrast phase microscope at 200× magnification. The percentage of motile sperm cells was estimated subjectively to the nearest 5% using an arbitrary scale of 0–100% [16].

Sperm samples were fixed and diluted 1:10 (*v*/*v*) in saline with 0.3% formaldehyde. A Bürker counting chamber was used to evaluate sperm concentration by contrast phase microscopy at 200× magnification, counting each sample in duplicate. To evaluate sperm morphology, a 10 µL sample drop was placed on a slide, covered with a 24 mm × 24 mm cover slip and morphology was evaluated by contrast phase microscopy at 1000× magnification (Leica DMR, Wetzlar, Germany). Two hundred spermatozoa per sample were counted and classified into sperm with normal morphology, sperm with proximal cytoplasmatic droplets, sperm with distal cytoplasmatic droplets, sperm with tail defects (folded and coiled tail), and sperm with other abnormal morphologies [16].

### 2.3. Photostimulation of Seminal Doses

Photo stimulation of the AI seminal doses was carried out using a commercial system (Maxipig, IUL SA, Barcelona, Spain) that illuminated the samples with red LED with a program of 10 min of light, followed by 10 min of darkness and finally another 10 min of light exposure [4]. The chamber maintained the temperature of the samples at 16 °C during the process.

### 2.4. Fertility Trial

The fertility study was conducted on two commercial farms in Murcia (Spain) using multiparous Iberian sows (parity range 2–11). After weaning, estrus was checked daily in the presence of a mature teaser boar. Occurrence of estrus was defined by the standing reflex in front of a boar (back pressure test) and reddening and swelling of the vulva. Post-cervical insemination took place 0 h after the diagnosis of estrus and was repeated 24 h later, using disposable post cervical AI catheters (Soft & Quick, Tecnovet SL., Barcelona, Spain) [17]. First and second artificial inseminations were performed with seminal doses with 3 × 10^9^ spermatozoa from the same ejaculate and the same treatment (LED or control). Inseminations were performed during the period 1 August 2018 to 31 December 2018.

Fertility was measured for every ejaculate as pregnancy rate (the percentage of sows with positive ultrasound diagnostic at day 30 after insemination to AI) and farrowing rate (the percentage of sows farrowing to AI). For each farrowed sow, the number of dead piglets (NBD) and piglets born alive (NBA) was recorded, and the sum was defined as the total number of piglets born (TB).

In study A, insemination doses from boars recovered on Monday and Wednesday were assigned to LED groups, and seminal doses recovered on Tuesday and Thursday were assigned to the control group. All the boars included in this study had samples that were photostimulated and some that were control. In study B, insemination doses from the same ejaculate every day were split into two groups (LED and control).

### 2.5. Statistical Analysis 

#### 2.5.1. Sample Size

We planned a study of independent cases and controls. Prior data indicated that the probability of exposure among controls was 0.87 [5]. If the true probability of exposure among cases was 0.91, we needed to study 960 cases and 960 controls to be able to reject the null hypothesis that the exposure rates for cases and controls are equal with a probability (power) of 0.8. The type I error probability associated with the test of this null hypothesis was 0.05. We used an uncorrected chi-squared statistic to evaluate this null hypothesis [18]. Finally, we studied 1413 controls and 1542 cases (LED), which was 50% higher than the calculated sample size.

#### 2.5.2. Data Analysis

The results are expressed as mean ± SEM and were analyzed by two-way ANOVA, considering the specific photostimulation treatment as a main variable and the other variables being the farm, parity, or boar. Pregnancy and farrowing data were modeled according to the binomial model of parameters and were analyzed by two-way ANOVA. When ANOVA revealed a significant effect, values were compared by Tukey’s test. Differences were considered statistically significant at *p* < 0.05.

Linear regression was used to further investigate relationships between litter size and measured semen parameters (Pearson’s correlation). Multiple regression was used to explore relationships between litter size and other factors.

## 3. Results

A total of 152 ejaculates from 38 boars were used in this study, with a volume of 179.34 ± 7.31; sperm concentration 591.11 ± 13.21 × 10^6^ sperm/mL; 106.01 ± 3.74 × 10^9^ sperm in the ejaculate; with 12.38 ± 0.81% morpho-anomalies, distributed as 1.84 ± 0.24% sperm with tail defects, 4.82 ± 0.36% with cytoplasmic proximal droplets, and 5.72 ± 0.55 with distal ones.

The photostimulation of the seminal doses had no effect on the reproductive performance (pregnancy rate, farrowing rate, litter size) in both farms. No differences were found between LED (L) and control (C) groups in either farm (*p* > 0.05) for pregnancy rate, duration of pregnancy, farrowing rate, and litter size (total, alive, and stillborn piglets). Farrowing rates in farm A were 88.77% (n = 383) for control and 89.57% (n = 441, *p =* 0.90) for the LED group. In farm B, the results were C: 90.53% (n = 1035) and L: 90.11% (n = 1101, *p =* 0.48). In farm A, total born piglets were 8.69 ± 0.11 for C and 8.71 ± 0.11 for L. In farm B, the results were 8.72 ± 0.07 for C and 8.70 ± 0.06 for L (*p =* 0.98) (Table 1).

Once we had not found any differences in the reproductive performance between LED and control groups, we studied other possible factors that could have affected the results, such as parity number, males, or sperm quality. No differences were found in fertility outcomes between LED and control groups for every group of parity. Parity influenced pregnancy and farrowing rate and litter size (TB and NBA) (Table 2).

The boar had a direct effect on farrowing and litter size, but not on pregnancy rate. No differences were found for the interaction between LED treatment and boar, which means that all boars followed the same pattern after photostimulation (Table 3). The analysis of the correlation between seminal parameters (n = 152) and fertility outcomes pointed out only a tendency (*p* = 0.09) for an inverse relationship (r =−0.14) between percentage of proximal cytoplasmic droplets and pregnancy rate. Finally, we applied a multivariate analysis to explore the factors affecting litter size (total piglets born). Parity and lactation days had a direct and positive effect on litter size (Table 4, *p* < 0.01), while total piglets born was inversely related to pregnancy length. The LED photostimulation had no effect, the same as for boar, insemination person, or the interval of weaning–estrus (Table 4, *p* > 0.05).

## 4. Discussion

In pigs, a red LED-based photostimulation procedure (10 min illumination, 10 darkness, and 10 illumination) is able to increase the whole boar sperm response to both the heat stress due to incubation at 37 °C for 90 min and the achievement of in vitro capacitation and subsequent progesterone-induced acrosome reaction [4]. However, in later studies, other groups did not find any improvement in motility, mitochondrial activity, nor viability after the same illumination procedure [19]. Finally, an increase in motility parameters after red LED stimulation was detected, but no alteration in viability, ROS production, or intracellular calcium [20].

The presence of opsins in the spermatozoa, the modification of the mitochondrial activity, and activation of plasma membrane receptors from the transient receptor potential (TRP) family of proteins have been suggested as possible mechanisms of action of the illumination on the sperm functionality [21,22,23,24,25]. However, in this study, these reported changes in the sperm parameters had no effect on the subsequent reproductive outcomes, measured in terms of farrowing rate and litter size under commercial conditions.

We hypothesize some different causes that could be related to these results. One hypothesis is that the possible improvement in the sperm parameters (not evaluated in this study) did not affect the reproductive outcomes because the insemination system was optimized (control group: 91.7% and 90.09% for pregnancy and farrowing rate). Under these circumstances, it is difficult to detect any improvement over these high values for reproductive parameters of the control group. Due to this, no relationship was found between sperm parameters and fertility as has previously been reported in other studies with more restrictive conditions [26,27,28]. On the other hand, the number of sperm in doses used in this study for post-cervical insemination was high (3 × 10^9^ sperm per dose). According to the compensatory theory supported by Amman [29,30], an increase in the number of viable spermatozoa in seminal doses will show an asymptotic curve in their relationship to fertility, limited by the reproductive potential of the male. It could be possible that these Iberian sows were near their maximum potential for reproductive characteristics, determined by their genetic values, and any additional improvement in seminal parameters would have no significant effect on them. Therefore, if this hypothesis is true, the possible improvement in reproductive performance must be related to female improvement by genetic selection more than seminal parameters.

Another alternative could be that the response of Duroc samples to LED photostimulation is different than other breeds. Some studies have pointed out differences between Duroc boars and other breeds in seminal parameters (volume, concentration, motility, viability, etc.) [31,32,33,34], testosterone concentration [35], and proportion of discarded semen samples in AI centers [36]. Interestingly, Tremoen et al. [37] reported a higher proportion of hyperactivated spermatozoa in Duroc samples than landrace at day 0 of storage. These differences could indicate different susceptibilities to the capacitation kinetic, which could be explored in further studies.

## 5. Conclusions

In conclusion, the application of photostimulation with red LED on the seminal doses from Duroc boars used for insemination of Iberian sows did not improve the reproductive outcomes under commercial conditions. The litter size parameters of the Iberian sows might be improved by genetic selection or improvement of the female conditions more readily than by improvements in the artificial insemination system that has already been optimized.

## Figures and Tables

**Table 1 animals-11-01656-t001:** Fertility outcomes of sows inseminated with seminal doses after photostimulation by LED or not (control). Data expressed as mean ± SEM.

Farm	Group	N	Pregnancy Rate	Farrowing Rate	NBA	NBD	TB
A	Control	383	92.43	88.77	8.35 ± 0.11	0.34 ± 0.03	8.69 ± 0.11
A	LED	441	93.20	89.57	8.35 ± 0.10	0.37 ± 0.03	8.71 ± 0.11
B	Control	1030	91.40	90.53	8.40 ± 0.07	0.32 ± 0.02	8.72 ± 0.07
B	LED	1101	91.47	90.11	8.42 ± 0.06	0.28 ± 0.02	8.70 ± 0.06
Total	Control	1413	91.72	90.09	8.8 ± 0.06	0.32 ± 0.02	8.71 ± 0.06
Total	LED	1542	91.96	89.95	8.40 ± 0.05	0.30 ± 0.02	8.70 ± 0.05
***p* Value**							
**Source of Variation**			**Pregnancy Rate**	**Farrowing Rate**	**NBA**	**NBD**	**TB**
Treatment			0.08	0.25	0.62	0.45	0.80
Farm			0.11	0.50	0.46	<0.01	0.04
Parity			0.43	0.05	<0.01	<0.01	<0.01
Farm Treatment			0.10	0.14	0.66	0.42	0.48
Parity Treatment			0.10	0.53	0.97	0.79	0.93
Farm Parity			0.43	0.11	0.77	0.10	0.92
Treatment * Farm * Parity			0.30	0.21	0.87	0.50	0.91

NBA: number of piglets born alive. NBD: number of piglets born dead (NBD). TB: total number of piglets born.

**Table 2 animals-11-01656-t002:** Reproductive outcome of Iberian sows according to parity.

Parity	N	Pregnancy Rate	Farrowing Rate	TB	NBA
2	403	90.82 ^ab^	87.34 ^a^	7.93 ± 0.11 ^a^	7.77 ± 0.11 ^a^
3	276	90.58 ^ab^	88.77 ^ab^	8.82 ± 0.13 ^b^	8.64 ± 0.13 ^b^
4	188	89.89 ^ab^	88.30 ^ab^	8.86 ± 0.16 ^b^	8.65 ± 0.15 ^b^
5	277	87.00 ^a^	85.56 ^a^	8.71 ± 0.13 ^b^	8.47 ± 0.13 ^b^
6	637	93.25 ^b^	92.62 ^b^	8.85 ± 0.08 ^b^	8.52 ± 0.08 ^b^
7	507	93.69 ^b^	91.52 ^b^	8.91 ± 0.09 ^b^	8.54 ± 0.09 ^b^
8	512	92.97 ^b^	91.02 ^b^	8.8 ± 0.09 ^b^	8.37 ± 0.09 ^b^
>8	155	92.26 ^ab^	90.32 ^ab^	8.66 ± 0.17 ^b^	8.15 ± 0.17 ^b^
***p* Value**					
**Source of Variation**		**Pregnancy Rate %**	**Farrowing Rate %**	**TB**	**NBA**
Treatment		0.40	0.76	0.98	0.75
Parity		0.03	0.02	<0.01	<0.01
Parity Treatment		0.03	0.47	0.80	0.99

^a, b^ in the same column represent differences at *p* < 0.05. TB: total number of piglets born. NBA: number of piglets born alive.

**Table 3 animals-11-01656-t003:** Effect of boar and red LED light treatment on reproductive parameters. ANOVA source of variation.

Source of Variation	Pregnancy Rate %	Farrowing Rate %	TB	NBA
Treatment	0.61	0.73	0.18	0.16
Boar	0.10	0.02	<0.01	0.02
Boar Treatment	0.39	0.53	0.50	0.54

TB: total number of piglets born. NBA: number of piglets born alive.

**Table 4 animals-11-01656-t004:** Multivariate analysis. Dependent variable: total piglets born N = 2.567 multiple R 0.12 squared multiple R 0.01.

Effect	Coefficient	Standard Error	*p*-Value
Parity	0.07	0.02	<0.01
Interval weaning–oestrus (days)	0.01	0.00	0.13
Lactation length (days)	0.04	0.01	<0.01
Pregancy length (days)	−0.08	0.03	<0.01
Boar	0.00	0.00	0.30
Insemination person	0.00	0.00	0.07
LED treatment	−0.12	0.07	0.09

## Data Availability

The data that support the findings of this study are available on request from the corresponding author.

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
