# Peer review of "Photo Stimulation of Seminal Doses with Red LED Light from Duroc Boars and Resultant Fertility in Iberian Sows"

_animals, 2021, doi:10.3390/ani11061656_

Round 1

Reviewer 1 Report

The premise of this work is to improve the reproductive performance of a pig breed that has very good seminal quality, further increasing it. In my opinion the treatment could even shorten the life of spermatozoa in the female genital tract so I am amazed that the results were not even worse. The work is well done in all its parts and the merit is to have demonstrated in the field, contradicting the bibliography, that the treatment on seminal material is useless. Regarding the interest of users in the subject it seems to me relative. I am very much in doubt about the suitability for publication, maybe it would be more suitable for a short comunication.

Author Response

The authors appreciate and acknowledge the work of the reviewer #1.

We agree with the reviewer that one of the benefits of this study is that it is developed in field conditions. Our opinion about the possible interest of readers of this article is not the same than reviewer´s one. The possibility to offer positive and negative results generated under field conditions and applying the scientific methodology to the practitioners, researchers and pig industry could be valuable.

Dear Reviewer  #1.  Thank you for your commentaries and suggestions. We hope now this manuscript will be suitable for publication in Animals.

Reviewer 2 Report

Studies have shown that LED-based photo stimulation of boar semen improves the reproductive performance of AI- sows although the mechanisms underlying this phenomenon has not been fully elucidated as yet. Under the experimental conditions used in this study, LED-based photo stimulation of boar semen did not seem to have any positive effects on the reproductive performance of  multiparous sows. The finding of this study reaffirm that LED-based photo stimulation could have varying effects on the fertility outcome of AI sows. The Reviewer suggests that some revisions are needed in the manuscript and it should be considered for publication.

Comments are as follows:

1. M&M

a) Give more information about the animal age. For L75-79 include the appropriate reference (Perhaps, Pérez-Patiño et al. 2018 Journal of Proteome Research, 17, 1065-1076).

b) Why used about 3 billion sperm for AI with the Soft & Quick set (L85; L98-99). When the sperm are deposited near the bicornuate uterus region, as in the case with the Soft & Quick post-CAI procedure, the AI dose should be less than two billion sperm. For frozen-thawed sperm some authors have used approx. 2 billion sperm. Please provide the appropriate reference (s) for the AI procedure.

c) Statistical analysis (L124-129). Some clarifications are needed in this section. The experimental design shows that there are four main factors (boar, treatment, farm and parity), and should consider to provide one table with the factorial ANOVA results for the analyzed parameters (farrowing rate, pregnancy, NBA, etc.). The Reviewer suggests that the interactions of the factors, perhaps only those with significant effect, could be presented in the table.

2. The reviewer suggests to replace ″increase″ with ″improve″ (L225). Also, should consider to remove the statement in L226-227- ″This is probably………of the control″. It is not a concluding statement.

3. Tables

a) Table 1 should show the ANOVA effects (all four of them), and the farrowing rate (without %) should be placed before the pregnancy rate (without %), etc. Explain in the text what are ″C″ and ″L″ referring to (L140).

b) Improve the caption of the tables, for examples, What is the control? What is ″Led″ referring to? Include means ± SEM.

c) Table 2 is similar to Figure 1, except the absence of the pregnancy rate and NBA. Should consider to present the results in a tabular form, similarly to Table 1, showing farrowing rate, pregnancy rate, NBA, NBD and TB. Figure 1 is not required

Author Response

The authors appreciate and acknowledge the work of the reviewer #2.

We will try to provide a point-by-point response to the reviewer’s comments

Comments are as follows:

  1. M&M
  2. a) Give more information about the animal age. For L75-79 include the appropriate reference (Perhaps, Pérez-Patiño et al. 2018 Journal of Proteome Research, 17, 1065-1076).

The age of the boars was in mean±sem of 20.10 ± 1.78 months old, ranged from 10 to 45 months. This information has been included in line 75.

The reference was added.

  1. b) Why used about 3 billion sperm for AI with the Soft & Quick set (L85; L98-99). When the sperm are deposited near the bicornuate uterus region, as in the case with the Soft & Quick post-CAI procedure, the AI dose should be less than two billion sperm. For frozen-thawed sperm some authors have used approx. 2 billion sperm. Please provide the appropriate reference (s) for the AI procedure.

We developed this study under field conditions, the normal procedure in these farms is the control group with this CAI procedure with high number of sperm per insemination.

The reference for CAI procedure was added.

  1. c) Statistical analysis (L124-129). Some clarifications are needed in this section. The experimental design shows that there are four main factors (boar, treatment, farm and parity), and should consider to provide one table with the factorial ANOVA results for the analyzed parameters (farrowing rate, pregnancy, NBA, etc.). The Reviewer suggests that the interactions of the factors, perhaps only those with significant effect, could be presented in the table.

We have followed your suggestions and prepared a new table with the factorial ANOVA. We have not included boar, because boars are used only in one farm.

Source of Variation

Pregnancy rate

Farrowing rate

NBA

NBD

TB

Treatment

0.08

0.25

0.62

0.45

0.80

Farm

0.11

0.50

0.46

<0.01

0.04

Parity

0.43

0.05

<0.01

<0.01

<0.01

Farm * treatment

0.10

0.14

0.66

0.42

0.48

Parity * treatment

0.10

0.53

0.97

0.79

0.93

Farm * Parity

0.43

0.11

0.77

0.10

0.92

Treatment * Farm * Parity

0.30

0.21

0.87

0.50

0.91

  1. The reviewer suggests to replace ″increase″ with ″improve″ (L225). Also, should consider to remove the statement in L226-227- ″This is probably………of the control″. It is not a concluding statement.

We follow your suggestions, thank you.

  1. Tables
  2. a) Table 1 should show the ANOVA effects (all four of them), and the farrowing rate (without %) should be placed before the pregnancy rate (without %), etc. Explain in the text what are ″C″ and ″L″ referring to (L140).

Table modified according to referee´s suggestions.

Source of Variation

Pregnancy rate

Farrowing rate

NBA

NBD

TB

Treatment

0.08

0.25

0.62

0.45

0.80

Farm

0.11

0.50

0.46

<0.01

0.04

Parity

0.43

0.05

<0.01

<0.01

<0.01

Farm * treatment

0.10

0.14

0.66

0.42

0.48

Parity * treatment

0.10

0.53

0.97

0.79

0.93

Farm * Parity

0.43

0.11

0.77

0.10

0.92

Treatment * Farm * Parity

0.30

0.21

0.87

0.50

0.91

We maintained first pregnancy rate because chronologically is before than farrowing.

  1. b) Improve the caption of the tables, for examples, What is the control? What is ″Led″ referring to? Include means ± SEM.
  2. c) Table 2 is similar to Figure 1, except the absence of the pregnancy rate and NBA. Should consider to present the results in a tabular form, similarly to Table 1, showing farrowing rate, pregnancy rate, NBA, NBD and TB. Figure 1 is not required

Figure 1 deleted.

Dear Reviewer  #2.  Thank you for your commentaries and suggestions. We hope now this manuscript will be suitable for publication in Animals.

Reviewer 3 Report

Review Manuscript animals-1227690, entitled „ Photo stimulation of seminal doses with red-led light from Duroc boars and resultant fertility in Iberian sows

The aim of the present study was to evaluate the possible beneficial effect on reproductive parameters of illuminating seminal doses with red-LED light from Duroc boars and Iberian females in their specific production system.

I have few points, which in my opinion should be explained:

What was the age of the boars? Was only one ejaculate collected from each boar (how many ejaculates from a boar)?

How the authors assessed sperm morphology, sperm concentration…. What were the methods used? When was this assessment carried out?

Was semen quality checked after photo stimulation of seminal doses?

The authors say that the litter size parameters of the Iberian sows might be improved by genetic selection or improvement of the female conditions. In my opinion, there is too little information on sperm parameters.

Author Response

The authors appreciate and acknowledge the work of the reviewer #3.

We will try to provide a point-by-point response to the reviewer’s comments 

  1. What was the age of the boars?

The age of the boars was in mean±sem of 20.10 ± 1.78 months old, ranged from 10 to 45 months. This information has been included in line 75.

  1. Was only one ejaculate collected from each boar (how many ejaculates from a boar)?

We detailed in section 3 Results (line 134) “A total of 152 ejaculates from 38 boars were used in this study”.

  1. How the authors assessed sperm morphology, sperm concentration…. What were the methods used? When was this assessment carried out?

Sperm Motility, Concentration and Morphology evaluation

For motility evaluation two subsamples were placed on warm glass slides (38°C) and examined under a contrast phase microscopy at 200 x magnification. The percentage of motile sperm cells was estimated subjectively to the nearest 5% using an arbitrary scale of 0–100%. (Gadea et al.  Reprod. Domes. Anim. 2004)

Sperm samples were fixed and diluted 1:10 (v/v) in saline with 0.3% formaldehyde. An Bürker counting chamber was used to evaluate sperm concentration by contrast phase microscopy at 200 x magnification, counting each sample in duplicate. To evaluate sperm morphology, a 10 µL-sample drop was placed on a slide, covered with a 24 mm x 24 mm cover slip and morphology were evaluated by contrast phase microscopy at 1000 x magnification (Leica DMR, Wetzlar, Germany). Two-hundred spermatozoa per sample were counted and classified into sperm with normal morphology, sperm with proximal cytoplasmatic droplets, sperm with distal cytoplasmatic droplets, sperm with tail defects (folded and coiled tail) and sperm with other abnormal morphologies. (Navarro-Serna, et al Animals 2021)

This evaluation was carried out immediately after sperm recovery.

This information has been added to the document.

  1. Was semen quality checked after photo stimulation of seminal doses?

The authors say that the litter size parameters of the Iberian sows might be improved by genetic selection or improvement of the female conditions. In my opinion, there is too little information on sperm parameters.

No, seminal quality was not evaluated after photo stimulation.

This study was developed under field conditions and following the photo stimulation of seminal doses according to the device instructions (Maxipig, IUL SA, Barcelona, Spain). The main objective was to evaluate the possible application in Iberian breeds to improve fertility and litter size, not to evaluate the effect on sperm parameters, because the field conditions did not lead the evaluation of sperm parameters with more accuracy technologies as CASA and flow cytometry.

Dear Reviewer  #3.  Thank you for your commentaries and suggestions. We hope now this manuscript will be suitable for publication in Animals.

Round 2

Reviewer 3 Report

The Authors made the necessary  improvements according to my suggestion. I have no other objections. In my opinion the manuscript can be published in ‘Animals’.